# OpenReview forum: "Reranker Helps, but Not Enough: Towards Strong Poisoning Attacks Against RAG"
_ICLR.cc/2026/Conference — Submitted to ICLR 2026_

### Official Review · Reviewer_ktoJ · 2025-10-22

**Soundness:** 3
**Presentation:** 3
**Contribution:** 2
**Rating:** 4
**Confidence:** 3

**Summary:**

The paper studies data poisoning attacks on Retrieval-Augmented Generation (RAG) systems and asks whether a reranker—a common module in realistic pipelines—can mitigate these attacks. The authors first show that adding a reranker indeed reduces the attack success rate of several existing poisoning methods, implying that prior work overestimates the vulnerability of RAG. To expose the remaining weaknesses, they propose a two-stage Prompt-Perturbation Poisoning Attack (P3A) that uses (1) rule-based prompt generation and (2) character-level perturbations guided by reranker gradients.  Experiments on multiple datasets and models demonstrate that P3A restores high attack success even with a reranker in place.

**Strengths:**

1. The paper addresses a practically important and underexplored problem: how reranking layers influence RAG’s robustness against poisoning. This perspective is novel and clarifies a gap in the current evaluation practice.

2. The paper is well written, easy to follow, and presents its experimental evidence clearly with good ablations and visualization.

**Weaknesses:**

1.  While the paper reports that rerankers lower attack success rates, it does not dig into the mechanism behind this effect. Conceptually, a reranker is still a learned retrieval model; why should it resist poisoning while the base retriever fails? A deeper analysis would make the finding more illuminating rather than purely empirical.

2. The P3A method assumes white-box access to the reranker in order to compute gradients and optimize perturbations. In realistic deployment, the attacker rarely knows which reranker a system uses. Therefore, the evaluation mainly reflects an upper bound on attack strength, not a feasible real-world threat. A discussion or experiment on black-box or transfer settings would be necessary to validate practicality.
3. Overall, I find this work meaningful and well-motivated, and I lean toward accepting it, provided the authors address the issues regarding insufficient explanation of why reranking helps and limited realism of the proposed attack.

**Questions:**

See Weaknesses.

---

> ### Author Response · Authors · 2025-11-18
> **Response to Reviewer ktoJ (Q1, Q2)**
>
> > Q1: Why should a reranker resist poisoning while the base retriever fails?
>
> We sincerely thank Reviewer ktoJ for this insightful question. We agree that understanding why rerankers reduce attack success rates is crucial, and we expand our analysis accordingly.
>
> Existing poisoning attacks typically **exploit simple lexical overlap or hotflip-based token manipulations** to ensure that the retriever pulls the malicious document into the top-k candidates. Their main focus is then to satisfy the generation condition. However, our study reveals a clear limitation of this approach. When we prepend the target question to each of the retriever’s top-J candidates and feed them into the reranker for scoring, we consistently observe that carefully crafted malicious texts are ranked much lower than expected. This shows that **prior attacks fail to satisfy the Retrieval and Reranking Condition**, which is required to successfully compromise a reranker-enhanced RAG system.
>
> The underlying reason is that, although a reranker is indeed a learned retrieval model, its architecture and role differ fundamentally from those of a dense retriever.
>
>  (1) **Rerankers perform full token–token cross-attention**, allowing explicit modeling of lexical alignment and long-range semantic interactions between the query and the document. As a result, poisoned texts that rely on surface-level similarity are more easily down-weighted.
>
>  (2) **Retrievers aim for high recall** and use approximate semantic embeddings, which are easier for poisoning attacks to exploit. In contrast, **rerankers aim for high precision** and perform a much more detailed, content-level relevance assessment. This fine-grained evaluation inherently filters out many artifacts introduced by existing poisoning strategies.
>
> > Q2:  A discussion or experiment on black-box or transfer settings would be necessary to validate practicality.
>
> We thank Reviewer ktoJ for this thoughtful comment. We agree that assuming full white-box access to the reranker may not always reflect realistic deployment scenarios.
>
> To clarify, P3A is a two-stage attack framework. The first stage uses rule-based prompt engineering to generate initial poisoned texts, while the second stage applies character-level perturbations. Importantly, **the first stage alone functions as a black-box variant, which we denote as P2A**. **As shown in Tables 2, 3, and 7, P2A already surpasses baseline methods in both vanilla RAG and reranker-enhanced RAG settings**, demonstrating that our approach remains highly effective even under black-box constraints.
>
> And **we have conducted transferability experiments (Table 6)** by training and testing P3A across different reranker architectures. The results show strong transferability, indicating that an attacker can use any open-source reranker as a proxy model to craft poisoned texts.
>
> --------
>
> We are grateful for the reviewer’s insightful comments and the time devoted to improving our manuscript. We hope our responses have addressed the concerns, and we welcome any further questions or suggestions to continue strengthening the work.

---

> ### Author Response · Authors · 2025-11-27
> **Request for Reviewer Feedback on Rebuttal**
>
> Dear Reviewer,
>
> I hope you are doing well. Thank you again for your time and helpful feedback.
>
> As the rebuttal period is approaching its end (**less than one week remaining**), we would like to make sure that we have fully addressed your concerns. If there are any remaining issues you would like us to consider, please feel free to let us know. Your suggestions are very valuable to us, and we are eager to further improve our work. We sincerely hope that our responses have helped clarify the points you raised and provided a more positive view of our contribution.
>
> Thank you once again for your time and effort in reviewing our paper.

---

### Official Review · Reviewer_m2V5 · 2025-10-30

**Soundness:** 3
**Presentation:** 2
**Contribution:** 3
**Rating:** 6
**Confidence:** 4

**Summary:**

This paper shows that rerankers inadvertently defend against existing targeted RAG poisoning attacks, and proposes a trageted attack P3A—a two-phase attack combining rule-based prompt engineering with character-level perturbations—that helps the attack survive in presence of rerankers with high attack success rates (>70% ASR).

**Strengths:**

1. Authors identify that existing targeted poisoning attacks on RAG are brittle when a re-ranker is present in the RAG system and consequently propose an attack that is effective in the presence of a re-ranker.
2. Comprehensive Evaluation with comparison against multiple baselines.

**Weaknesses:**

1. Description of the methodology is unclear.
2. No description on why existing techniques such HotFlip, GCG or other search based techniques can't be used.
3. Does this attack translate to production RAG systems.

**Questions:**

1. Character Level Perturbation: How are the initial set of n candidates chosen before PGD. Why go through two step process as described. Why can't we use existing techniques such as Gradient based techniques: HotFlip, GCG or Search based techniques eariler used for jailbreaking, to be repurposed in this scenario.

2. It would be good to test out your attack by connecting the RAG database with your poisoned documents to see if you can get non-zero attack success against production RAG systems to show real world impact.

---

> ### Author Response · Authors · 2025-11-18
> **Response to Reviewer m2V5 (Q1, Q2)**
>
> > Q1: Character level perturbation.
>
> We thank Reviewer m2V5 for this thoughtful question. We are glad to clarify the motivation behind our two-stage design and the choice of candidate initialization before PGD.
>
> Our character-level optimization must address two challenges: **(1) identifying where perturbations meaningfully influence the reranker score, and (2) optimizing how to modify those positions.**
>
> Stage 1 addresses the first challenge by using the reranker score to locate the most influential positions $\mathcal{Z}^*$. By masking each candidate position with a space and measuring its score impact, we can reliably identify vulnerable locations. This step reduces the search space and focuses on positions that matter most for the reranker.
>
> Stage 2 then generates $n$ perturbed candidates by applying simple character-level edits (insert, delete, or substitute with a random character) to the positions in $\mathcal{Z}^*$. These candidates serve as anchors for PGD. This design allows gradients to guide us toward more promising perturbations while still respecting the discrete nature of character-level edits.
>
> Regarding the use of existing attack methods such as HotFlip, GCG, or prior jailbreak search techniques: these approaches operate primarily at the token level. In contrast, **our setting requires fine-grained character-level modifications, which are essential for maintaining stealthiness and readability**. Token-level gradient methods are not directly compatible with this granularity, and applying them would either produce unnatural edits or lead to excessive disturbance, which is easily detectable.
>
> Moreover, **as shown in our evaluation of textual modifications (Table 4), the character-level perturbations modify only about 1% of the text while still delivering substantially stronger poisoning effects**. This level of minimal, localized editing is not achievable with token-level methods, which typically introduce larger and more noticeable changes. These results underscore the necessity and advantage of character-level editing for achieving both high stealth and high attack success.
>
> > Q2: Production RAG systems.
>
> We thank Reviewer m2V5 for this constructive suggestion. Demonstrating real-world impact is indeed important.
>
> In our study, **we follow prior work and conduct extensive experiments on NQ, MS-MARCO, and HotpotQA[1-3]**, whose corpora are constructed from large-scale, real-world sources such as Wikipedia and web documents. The databases typically contain 2–10M passages, capturing the diversity and noise characteristics commonly seen in practical RAG deployments. Evaluating our attack under these conditions allows us to approximate real production-level RAG environments.
>
> To further address the reviewer’s concern, **we also include additional experiments on the FEVER dataset[4], which focuses on fact verification, a task that differs from traditional QA**. This extension enriches our evaluation by introducing a new task and corpus. Our results on FEVER show that P3A continues to outperform existing baselines, demonstrating that our method remains effective not only on smaller QA datasets but also scales well to larger and more diverse retrieval scenarios.
>
> |                  | Acc   | ASR   |
> |------------------|-------|-------|
> | Prompt Injection | 48.6  | 40.2  |
> | Disinformation   | 55.6  | 35.4  |
> | PoisonedRAG-B    | 60.0  | 34.6  |
> | PoisonedRAG-W    | 55.2  | 32.2  |
> | CorruptRAG-AS    | 35.2  | 49.0  |
> | CorruptRAG-AK    | 40.8  | 45.6  |
> | P2A            | 28.0  | 49.6  |
> | P3A            | **26.8**  | **52.2**  |
>
> **Table 1. Performance of different attack methods on the FEVER dataset using a MiniLM reranker and Llama3-8B.**
>
> We fully agree that evaluating attacks on a fully deployed, end-to-end RAG system would further strengthen the practical relevance. We will incorporate this direction into future work.
>
> -----
>
> Thank you again for the helpful feedback and thoughtful guidance. We hope our detailed responses successfully resolve the reviewer’s concerns. If there are any remaining uncertainties, we would be happy to clarify them.
>
> [1] PoisonedRAG: Knowledge Corruption Attacks to Retrieval-Augmented Generation of Large Language Models. USENIX Security Symposium 2025
>
> [2] Machine Against the RAG: Jamming Retrieval-Augmented Generation with Blocker Documents. USENIX Security Symposium 2025
>
> [3] Practical Poisoning Attacks against Retrieval-Augmented Generation.
>
> [4] Poisoning Retrieval Corpora by Injecting Adversarial Passages. EMNLP 2023

---

> ### Author Response · Authors · 2025-11-27
> **Request for Reviewer Feedback on Rebuttal**
>
> Dear Reviewer,
>
> I hope you are doing well. Thank you again for your time and helpful feedback.
>
> As the rebuttal period is approaching its end (**less than one week remaining**), we would like to make sure that we have fully addressed your concerns. If there are any remaining issues you would like us to consider, please feel free to let us know. Your suggestions are very valuable to us, and we are eager to further improve our work. We sincerely hope that our responses have helped clarify the points you raised and provided a more positive view of our contribution.
>
> Thank you once again for your time and effort in reviewing our paper.

---

### Official Review · Reviewer_6xRo · 2025-10-31

**Soundness:** 2
**Presentation:** 3
**Contribution:** 2
**Rating:** 2
**Confidence:** 5

**Summary:**

This paper presents the Prompt-Perturbation Poisoning Attack (P3A) framework, which integrates a rule-based prompt generation phase guided by reranker-oriented prompt engineering with a character-level perturbation phase that refines texts to improve their ranking, thereby enabling effective attacks against RAG systems.

**Strengths:**

1. The paper proposes a new attack method targeting RAG systems.

2. Experimental results demonstrate the effectiveness of the proposed approach.

**Weaknesses:**

1. The threat model is unrealistic.

2. Injecting five poisoned documents into the system is impractical.

3. The paper does not consider recent defense methods.

**Questions:**

1. The proposed P3A attack relies on a white-box assumption where the attacker has full access to the reranker’s parameters and gradients. However, in practical RAG systems, rerankers are usually proprietary or access-restricted. Therefore, this assumption is unrealistic, and the authors should evaluate the attack under a more practical black-box setting.

2. In the experiments, the attack injects five malicious documents per query into the corpus. This setup is impractical because, as shown in [a], the number of truly relevant texts among the top-5 retrieved documents per query is typically fewer than five (e.g., in the NQ dataset). Consequently, injecting five malicious documents means that the number of poisoned texts exceeds the number of relevant ones, trivially increasing the attack success rate. The authors should instead adopt a realistic constraint, allowing only one poisoned document per query.

3. The paper evaluates only basic defenses such as perplexity-based filtering and query paraphrasing, while neglecting more advanced and robust defenses proposed in recent studies [b][c][d].

4. The current experiments are conducted only on small-scale datasets such as NQ, MS-MARCO, and HotpotQA. To validate the generalizability and scalability of the proposed method, the authors should further evaluate its performance on large-scale datasets.


[a] Practical Poisoning Attacks against Retrieval-Augmented Generation.

[b] TrustRAG: Enhancing Robustness and Trustworthiness in RAG.

[c] Certifiably robust rag against retrieval corruption.

[d] Who Taught the Lie? Responsibility Attribution for Poisoned Knowledge in Retrieval-Augmented Generation. In IEEE Symposium on Security and Privacy, 2026.

---

> ### Author Response · Authors · 2025-11-18
> **Response to Reviewer 6xRo (Q1, Q2)**
>
> > Q1: Black-box setting.
>
> We sincerely thank Reviewer 6xRo for the insightful comment.
>
> We clarify that our proposed P3A attack is a two-stage framework, where the first stage employs rule-based prompt engineering to construct initial poisoned texts, followed by character-level perturbation. Importantly, **the first stage alone constitutes a black-box variant, which we term P2A**. As shown in Table 2, 3, 7, P2A already outperforms all baselines in both vanilla RAG and reranker-enhanced RAG settings, demonstrating that our method remains effective even when the attacker does not access model parameters or gradients.
>
> Moreover, **we have conducted transferability experiments in the main text (Table 6)**. By training and testing P3A across different reranker architectures, we observe strong transferability. This suggests that an attacker can rely on an open-source reranker as a proxy model to craft poisoned texts, further alleviating the need for white-box access to the target system.
>
> > Q2: Impact of the number of poisoned documents.
>
> We thank Reviewer 6xRo for this valuable observation. Our default setting of injecting five malicious documents per query follows the widely adopted experimental protocol in PoisonedRAG[1].
>
> Importantly, **we have conducted an analysis on the impact of the number of poisoned documents, as presented in Figures 4 and 6**. The results show that P3A remains highly effective even when a single poisoned document is injected, and notably, its performance surpasses baseline attacks that inject five poisoned documents. This demonstrates that P3A does not rely on injecting a large number of malicious texts.
>
> To further clarify this point, we present **quantitative results for the case of one poisoned document per query**. These additions highlight that P3A remains effective even under this more realistic constraint.
>
> |                  | NQ    |       | MS-MARCO |       | HotpotQA |       |
> |------------------|-------|-------|----------|-------|----------|-------|
> | Method           | Acc   | ASR   | Acc      | ASR   | Acc      | ASR   |
> | Prompt Injection | 37.4  | 2.4   | 32.0     | 4.8   | 39.6     | 10.0  |
> | Disinformation   | 32.2  | 24.6  | 27.8     | 20.4  | 36.0     | 25.4  |
> | PoisonedRAG-B    | 32.6  | 27.0  | 30.2     | 18.4  | 34.4     | 26.0  |
> | PoisonedRAG-W    | 33.6  | 13.4  | 29.8     | 9.8   | 32.0     | 18.2  |
> | CorruptRAG-AS    | 36.4  | 3.4   | 32.0     | 5.0   | 38.8     | 9.8   |
> | CorruptRAG-AK    | 33.2  | 15.6  | 32.0     | 5.0   | 38.2     | 12.2  |
> | P2A              | 26.0  | 47.8  | 28.0     | 24.8  | **26.0**     | **54.8**  |
> | P3A              | **24.6**  | **58.6**  | **26.8**     | **31.6**  | 26.2     | 54.2  |
>
> **Table 1. Performance of different attack methods using a MiniLM reranker and Llama3-8B under the setting of one poisoned document per query.**

---

> ### Author Response · Authors · 2025-11-18
> **Response to Reviewer 6xRo (Q3)**
>
> > Q3: Advanced defense.
>
> We thank Reviewer 6xRo for the valuable suggestion. We agree that evaluating the advanced defenses would provide a deeper understanding of the attack’s effectiveness.
>
> Given that advanced defenses tend to have high computational costs, we initially focused on more commonly used defenses, such as **perplexity-based filtering and query paraphrasing, which are efficient and widely implemented in practice[1-3]**. In response to the reviewer’s concern, we include additional experiments using the defenses from RobustRAG and TrustRAG, following their experimental setups[4, 5].
>
> As shown in the results below, these defenses provide some level of protection; however, our proposed methods continue to **exhibit substantial attack success rates**. Additionally, **the defense runtimes are notably longer**, with the time (minutes) per 100 queries as follows. The extended runtimes can significantly impact system efficiency, especially in large-scale deployments.
>
>
> |          |                | w/o Defense |       | RobustRAG |       | TrustRAG |       |
> |----------|----------------|-------------|-------|-----------|-------|----------|-------|
> | NQ       | Disinformation | 30.2        | 27.6  | 30.4      | 24.0  | 42.6     | 19.8  |
> |          | PoisonedRAG-B  | 28.6        | 34.8  | 31.8      | 28.8  | 41.4     | 21.8  |
> |          | P2A            | 15.8        | 63.0  | 16.2      | 62.4  | 31.4     | 36.0  |
> |          | P3A            | 10.4        | 72.2  | 15.6      | 69.2  | 26.0     | 48.0  |
> | MS-MARCO | Disinformation | 27.6        | 25.2  | 28.8      | 15.6  | 32.2     | 15.4  |
> |          | PoisonedRAG-B  | 28.8        | 24.4  | 26.4      | 16.0  | 31.2     | 16.2  |
> |          | P2A            | 21.2        | 40.6  | 22.2      | 34.4  | 29.8     | 20.0  |
> |          | P3A            | 15.6        | 53.0  | 18.4      | 41.8  | 25.2     | 27.8  |
> | HotpotQA | Disinformation | 33.0        | 34.4  | 26.0      | 24.4  | 40.6     | 32.4  |
> |          | PoisonedRAG-B  | 30.2        | 35.0  | 24.4      | 31.8  | 39.6     | 30.0  |
> |          | P2A            | 13.4        | 73.2  | 11.4      | 71.4  | 22.6     | 52.8  |
> |          | P3A            | 13.0        | 72.8  | 10.6      | 74.6  | 21.0     | 55.8  |
>
> **Table 2. Results under different advanced defense methods using a MiniLM reranker and Llama3-8B.**
>
> |             | Time  |
> |-------------|-------|
> | w/o Defense | 3:23  |
> | RobustRAG   | 25:20 |
> | TrustRAG    | 16:45 |
>
> **Table 3. Runtime (minutes) per 100 queries for different defense methods.**

---

> ### Author Response · Authors · 2025-11-18
> **Response to Reviewer 6xRo (Q4)**
>
> > Q4: Large-scale datasets.
>
> We thank Reviewer 6xRo for this valuable suggestion. We agree that evaluating the performance on larger-scale datasets is essential to demonstrate the generalizability and scalability of our method.
>
> The datasets we used, **NQ, MS-MARCO, and HotpotQA, are commonly employed in RAG poisoning attack research[1-5]**, and their corpora, sourced from the web, provide a diverse set of retrieval conditions. These datasets already offer a good indication of the generalizability to real-world tasks.
>
> To address the reviewer’s concern, **we include additional experiments on the FEVER dataset[6]**, which focuses on fact verification, a task that differs from traditional QA. This extension enriches our evaluation by introducing a new task and corpus. The FEVER corpus consists of >5M documents, larger than NQ and HotpotQA. Our results on FEVER demonstrate that our method is not only effective on smaller datasets but also scales well to larger and more diverse tasks.
>
> |                  | Acc   | ASR   |
> |------------------|-------|-------|
> | Prompt Injection | 48.6  | 40.2  |
> | Disinformation   | 55.6  | 35.4  |
> | PoisonedRAG-B    | 60.0  | 34.6  |
> | PoisonedRAG-W    | 55.2  | 32.2  |
> | CorruptRAG-AS    | 35.2  | 49.0  |
> | CorruptRAG-AK    | 40.8  | 45.6  |
> | P2A            | 28.0  | 49.6  |
> | P3A            | **26.8**  | **52.2**  |
>
> **Table 4. Performance of different attack methods on the FEVER dataset using a MiniLM reranker and Llama3-8B.**
>
> > Reminder of broader contribution.
>
> We would like to highlight that the main contribution of our work goes beyond introducing P3A and P2A. **Our study uncovers a fundamental limitation in existing RAG poisoning attacks**: rerankers fine-tuned solely on benign data already provide a surprisingly strong “free-lunch” defense, substantially reducing the effectiveness of prior methods. This observation exposes an important weakness in current red-teaming practices and underscores the need for more robust evaluation frameworks.
>
> ------------------
>
> We sincerely thank the reviewer for the thoughtful and constructive feedback. We hope that our responses adequately address the concerns raised and clarify the contributions of our work. Please feel free to let us know if any issues remain—we would be glad to further elaborate.
>
> [1] PoisonedRAG: Knowledge Corruption Attacks to Retrieval-Augmented Generation of Large Language Models. USENIX Security Symposium 2025
>
> [2] Machine Against the RAG: Jamming Retrieval-Augmented Generation with Blocker Documents. USENIX Security Symposium 2025
>
> [3] Practical Poisoning Attacks against Retrieval-Augmented Generation.
>
> [4] Certifiably robust rag against retrieval corruption.
>
> [5] TrustRAG: Enhancing Robustness and Trustworthiness in RAG.
>
> [6] Poisoning Retrieval Corpora by Injecting Adversarial Passages. EMNLP 2023

---

> ### Author Response · Authors · 2025-11-27
> **Request for Reviewer Feedback on Rebuttal**
>
> Dear Reviewer,
>
> I hope you are doing well. Thank you again for your time and helpful feedback.
>
> As the rebuttal period is approaching its end (**less than one week remaining**), we would like to make sure that we have fully addressed your concerns. If there are any remaining issues you would like us to consider, please feel free to let us know. Your suggestions are very valuable to us, and we are eager to further improve our work. We sincerely hope that our responses have helped clarify the points you raised and provided a more positive view of our contribution.
>
> Thank you once again for your time and effort in reviewing our paper.

---

> > ### Comment · Reviewer_6xRo · 2025-11-27
> >
> > All my concerns have been addressed, so I have decided to increase my score.

---

> > > ### Author Response · Authors · 2025-11-28
> > >
> > > Thank you very much for reconsidering your evaluation. Your comments have been very helpful in improving the paper. We truly appreciate your thoughtful feedback and recognition of our work.

---

### Official Review · Reviewer_FgvL · 2025-11-01

**Soundness:** 2
**Presentation:** 2
**Contribution:** 3
**Rating:** 4
**Confidence:** 3

**Summary:**

This paper explores the vulnerability of RAG)systems to data poisoning and shows that while rerankers, offer a surprising “free defense” against existing attacks, they are not sufficient. To expose these weaknesses, the authors propose the Prompt-Perturbation Poisoning Attack (P3A), which first uses prompt engineering to create realistic, authoritative poisoned documents and then applies tiny character-level tweaks (about 1% of the text) to boost their ranking while keeping them natural. Experiments across multiple datasets and models demonstrate that P3A significantly outperforms prior methods, effectively compromising even reranker-enhanced RAG systems and transferring well to vanilla ones. The study concludes that rerankers help but cannot fully defend RAG pipelines, highlighting the urgent need for stronger, more adaptive defenses.

**Strengths:**

1. Proposes a smart new attack, P3A, that mixes prompts and tiny text tweaks.
2. Works in both black-box and white-box settings.
3. Tested on many datasets, rerankers, and LLMs.

**Weaknesses:**

1. A key limitation of P3A is that its full-power version requires white-box access to the reranker, the character-level PGD and position-selection steps depend on seeing reranker scores/gradients, so the fine-grained perturbation phase can’t be executed in a strict black-box setting. The paper does offer a black-box variant (P2A) that relies only on rule-based prompt engineering to produce “reranker-friendly” poisoned texts, and that variant performs well in experiments, but it generally lacks the precision of white-box P3A. In practice an attacker might try to optimize against a publicly available or proxy reranker and hope the poisoned samples transfer to the target system; this proxy to target transfer often works but is an empirical assumption that can fail when architectures, pre-processing, or retrieval configurations differ.

2. The paper provides limited concrete mitigation strategies or operational deployment recommendations; a more thorough discussion of defenses, detection tradeoffs, and ethical considerations would increase practical impact.

3. Experiments focus on three QA datasets and targeted factoid queries; it is unclear how the attack generalizes to other RAG applications (multi-turn dialogue, summarization, multimodal retrieval, or knowledge bases).

4. The paper mostly relies on injecting multiple poisoned docs (they run with 5), so its big wins may overstate real-world risk, flooding a corpus is noisier and easier to spot than a single stealthy page. They show P3A can work with one doc, but I’d like to see more results and discussion about the minimum poisons needed and how detectable bulk injections are.

**Questions:**

N/A

---

> ### Author Response · Authors · 2025-11-18
> **Response to Reviewer FgvL (Q1, Q2)**
>
> > Q1: Limitation of white-box access.
>
> We thank Reviewer FgvL for the insightful comment. To ensure we fully address your concern, could you please clarify whether your main question is about the performance of our black-box variant, P2A, or the transferability of the P3A method in proxy-to-target scenarios? We would appreciate any **further clarification on your specific concerns**.
>
> We would like to **emphasize that our black-box variant, P2A, performs effectively as demonstrated in Tables 2, 3, and 7**. P2A outperforms all baselines in both vanilla RAG and reranker-enhanced RAG settings, showing that it remains effective even under black-box constraints.
>
> **We have conducted transferability experiments (Table 6)**, where we train and test P3A on different reranker architectures. The results show strong transferability across models, which suggests that an attacker could rely on a publicly available reranker to craft poisoned texts.
>
> Finally, we would like to emphasize that the key contribution of our paper is not just the proposal of P3A and P2A, but **the identification of a significant gap in existing RAG poisoning attacks**. We show that rerankers fine-tuned on benign data offer a “free lunch” defense, effectively mitigating traditional poisoning strategies. This reveals a major limitation in the current red-teaming effort.
>
> > Q2: Limited concrete mitigation strategies.
>
> **In line with prior studies[1-3], we evaluated commonly used defenses (Figure 3, Table 5)** such as perplexity-based filtering and query paraphrasing, which are widely deployed in practice due to their low computational cost. Our experiments show that these defenses are not effective against the P3A attack.
>
> **We also conducted experiments on textual modifications(Table 4)**, where we measured the edit distance of the poisoned texts. The results indicate that P3A introduces minimal changes to the texts, preserving their readability and making the modifications largely undetectable.
>
> Additionally, we have **included results for advanced defense methods**, RobustRAG[4] and TrustRAG[5]. While these defenses offer some protection, they come with significant computational overhead, and P3A continues to achieve high attack success rates despite these advanced defenses.
>
>
> |          |                | w/o Defense |       | RobustRAG |       | TrustRAG |       |
> |----------|----------------|-------------|-------|-----------|-------|----------|-------|
> | NQ       | Disinformation | 30.2        | 27.6  | 30.4      | 24.0  | 42.6     | 19.8  |
> |          | PoisonedRAG-B  | 28.6        | 34.8  | 31.8      | 28.8  | 41.4     | 21.8  |
> |          | P2A            | 15.8        | 63.0  | 16.2      | 62.4  | 31.4     | 36.0  |
> |          | P3A            | 10.4        | 72.2  | 15.6      | 69.2  | 26.0     | 48.0  |
> | MS-MARCO | Disinformation | 27.6        | 25.2  | 28.8      | 15.6  | 32.2     | 15.4  |
> |          | PoisonedRAG-B  | 28.8        | 24.4  | 26.4      | 16.0  | 31.2     | 16.2  |
> |          | P2A            | 21.2        | 40.6  | 22.2      | 34.4  | 29.8     | 20.0  |
> |          | P3A            | 15.6        | 53.0  | 18.4      | 41.8  | 25.2     | 27.8  |
> | HotpotQA | Disinformation | 33.0        | 34.4  | 26.0      | 24.4  | 40.6     | 32.4  |
> |          | PoisonedRAG-B  | 30.2        | 35.0  | 24.4      | 31.8  | 39.6     | 30.0  |
> |          | P2A            | 13.4        | 73.2  | 11.4      | 71.4  | 22.6     | 52.8  |
> |          | P3A            | 13.0        | 72.8  | 10.6      | 74.6  | 21.0     | 55.8  |
>
> **Table 1. Results under different advanced defense methods using a MiniLM reranker and Llama3-8B.**
>
> |             | Time  |
> |-------------|-------|
> | w/o Defense | 3:23  |
> | RobustRAG   | 25:20 |
> | TrustRAG    | 16:45 |
>
> **Table 2. Runtime (minutes) per 100 queries for different defense methods.**

---

> ### Author Response · Authors · 2025-11-18
> **Response to Reviewer FgvL (Q3, Q4)**
>
> > Q3: How the attack generalizes to other RAG applications.
>
> We thank Reviewer FgvL for the thoughtful comment. The datasets, **NQ, MS-MARCO, and HotpotQA, are widely employed in previous RAG poisoning attack studies[1-6] and represent diverse tasks**. These datasets cover a range of query types, including factoid questions, multi-hop reasoning, and long-answer extraction, reflecting a broad spectrum of real-world applications.
>
> To address the reviewer’s concern regarding the generalization of our attack, we have also **evaluated our method on the FEVER dataset[6], which focuses on fact verification**. This task introduces a different challenge compared to traditional QA, expanding the scope of our experiments. Our results show that P3A continues to outperform baseline methods on FEVER, demonstrating the attack's effectiveness across varied tasks.
>
> In future work, we plan to focus on other RAG applications, such as multimodal retrieval and multi-turn dialogue, but these are beyond the scope of the current paper.
>
> |                  | Acc   | ASR   |
> |------------------|-------|-------|
> | Prompt Injection | 48.6  | 40.2  |
> | Disinformation   | 55.6  | 35.4  |
> | PoisonedRAG-B    | 60.0  | 34.6  |
> | PoisonedRAG-W    | 55.2  | 32.2  |
> | CorruptRAG-AS    | 35.2  | 49.0  |
> | CorruptRAG-AK    | 40.8  | 45.6  |
> | P2A            | 28.0  | 49.6  |
> | P3A            | **26.8**  | **52.2**  |
>
> **Table 3. Performance of different attack methods on the FEVER dataset using a MiniLM reranker and Llama3-8B.**
>
> > Q4: Impact of the number of poisoned documents.
>
> We thank Reviewer FgvL for the thoughtful comment. We agree that in practical scenarios, injecting multiple poisoned documents could be more easily detected and may not accurately represent the stealthiness of a real-world attack.
>
> Our default setting of injecting five malicious documents per query follows the established protocol in PoisonedRAG[1]. Moreover, **we have conducted a detailed analysis on the impact of the number of poisoned documents, as shown in Figure 4 and 6**. These results demonstrate that P3A remains effective even with a single poisoned document and, in fact, outperforms baseline methods that inject five poisoned documents.
>
> To further clarify this point, **we present quantitative results for the case of one poisoned document per query**. These additions highlight that P3A remains effective even under this more realistic constraint.
>
> |                  | NQ    |       | MS-MARCO |       | HotpotQA |       |
> |------------------|-------|-------|----------|-------|----------|-------|
> | Method           | Acc   | ASR   | Acc      | ASR   | Acc      | ASR   |
> | Prompt Injection | 37.4  | 2.4   | 32.0     | 4.8   | 39.6     | 10.0  |
> | Disinformation   | 32.2  | 24.6  | 27.8     | 20.4  | 36.0     | 25.4  |
> | PoisonedRAG-B    | 32.6  | 27.0  | 30.2     | 18.4  | 34.4     | 26.0  |
> | PoisonedRAG-W    | 33.6  | 13.4  | 29.8     | 9.8   | 32.0     | 18.2  |
> | CorruptRAG-AS    | 36.4  | 3.4   | 32.0     | 5.0   | 38.8     | 9.8   |
> | CorruptRAG-AK    | 33.2  | 15.6  | 32.0     | 5.0   | 38.2     | 12.2  |
> | P2A              | 26.0  | 47.8  | 28.0     | 24.8  | **26.0**     | **54.8**  |
> | P3A              | **24.6**  | **58.6**  | **26.8**     | **31.6**  | 26.2     | 54.2  |
>
> **Table 4. Performance of different attack methods using a MiniLM reranker and Llama3-8B under the setting of one poisoned document per query.**
>
> -----------
>
> We truly appreciate the reviewer’s careful evaluation and valuable suggestions. We hope our explanations resolve the questions raised. Should any part require further clarification, we are more than willing to provide additional details.
>
>
> [1] PoisonedRAG: Knowledge Corruption Attacks to Retrieval-Augmented Generation of Large Language Models. USENIX Security Symposium 2025
>
> [2] Machine Against the RAG: Jamming Retrieval-Augmented Generation with Blocker Documents. USENIX Security Symposium 2025
>
> [3] Practical Poisoning Attacks against Retrieval-Augmented Generation.
>
> [4] Certifiably robust rag against retrieval corruption.
>
> [5] TrustRAG: Enhancing Robustness and Trustworthiness in RAG.
>
> [6] Poisoning Retrieval Corpora by Injecting Adversarial Passages. EMNLP 2023

---

> ### Author Response · Authors · 2025-11-27
> **Request for Reviewer Feedback on Rebuttal**
>
> Dear Reviewer,
>
> I hope you are doing well. Thank you again for your time and helpful feedback.
>
> As the rebuttal period is approaching its end (**less than one week remaining**), we would like to make sure that we have fully addressed your concerns. If there are any remaining issues you would like us to consider, please feel free to let us know. Your suggestions are very valuable to us, and we are eager to further improve our work. We sincerely hope that our responses have helped clarify the points you raised and provided a more positive view of our contribution.
>
> Thank you once again for your time and effort in reviewing our paper.

---

### Meta-Review · Area_Chair_KNNx · 2026-01-05

**Summary:**

The paper observes that rerankers provide implicit defense against existing RAG poisoning attacks, then proposes P3A to bypass this defense via prompt-based generation and character-level perturbation. Experiments span NQ, MS-MARCO, HotpotQA, and FEVER with multiple rerankers and LLMs.

**Reviewer Concerns:**

Addressed: Authors provided P2A as black-box variant, cross-reranker transfer results, and single-document poisoning experiments. Additional defenses (RobustRAG, TrustRAG) were evaluated.

Outstanding: Two issues remain insufficiently resolved.

First, the threat model realism. The default 5-document poisoning assumption was questioned by 6xRo as potentially trivializing attack success. While authors added 1-doc results, ktoJ (rating 4) explicitly noted concerns about whether the attack scenario reflects realistic deployment conditions. The white-box assumption for P3A's character-level optimization requires gradient access to the reranker, which limits practical applicability despite transfer experiments.

Second, the mechanistic understanding is shallow. ktoJ raised that the explanation for why rerankers resist poisoning (cross-attention vs embedding approximation) remains surface-level and lacks rigorous analysis. This reviewer did not respond post-rebuttal, leaving the concern unresolved rather than addressed.

Additionally, two of four reviewers rated the paper at 4 (marginally below threshold). FgvL stated the contribution is "marginally below" acceptance standard, and ktoJ expressed the work is borderline with unresolved concerns about realism.

**Reviewer Scores:**

6xRo (2→6, Conf 5): Updated to above-threshold after rebuttal.
m2V5 (6, Conf 4): Above-threshold; asked for clearer method description and justification vs token-level attacks; suggested more “production” validation.
ktoJ (4, Conf 3): Borderline; requested deeper mechanism analysis for why rerankers resist poisoning and stronger realism justification (white-box/transfer). No post-rebuttal update.
FgvL (4, Conf 3): Borderline. No post-rebuttal update.

---

### Decision · Program_Chairs · 2026-01-26

Reject